# Assessment of Groundwater Quality and Human Health Risk (HHR) Evaluation of Nitrate in the Central-Western Guanzhong Basin, China

**DOI:** 10.3390/ijerph16214246

**Published:** 2019-11-01

**Authors:** Qiying Zhang, Panpan Xu, Hui Qian

**Affiliations:** 1School of Environmental Science and Engineering, Chang’an University, Xi’an 710054, China; zhangqiying@chd.edu.cn (Q.Z.); 2017029001@chd.edu.cn (P.X.); 2Key Laboratory of Subsurface Hydrology and Ecological Effects in Arid Region of the Ministry of Education, Chang’an University, Xi’an 710054, China

**Keywords:** groundwater quality, nitrate contamination, health risk assessment, water quality index, Guanzhong Basin

## Abstract

To investigate the quality of domestic groundwater and assess its risk to inhabitants of the Guanzhong Basin, China, 191 groundwater samples were collected to analyze major ions, nitrate, pH, total dissolved solids (TDS), total hardness (TH), and electrical conductivity (EC). The physiochemical parameters, hydrochemical facies, and sources of major ions were analyzed using Durov diagrams, bivariate diagrams, and chloro-alkaline indices (CAI-I and CAI-II). The suitability of groundwater for drinking, the nitrate distribution, and human health risk (HHR) for different age groups were evaluated. The results showed that the relative abundance of cations in the groundwater samples was K^+^+Na^+^ > Ca^2+^ > Mg^2+^, while that of anions was HCO_3_^−^ > SO_4_^2−^ > Cl^−^ > NO_3_^−^. Groundwater samples mainly contained HCO_3_-Na and HCO_3_-Ca, which were introduced mainly by rock weathering and ion exchange. The groundwater in the Guanzhong Basin contained mainly good and medium water, and the groundwater in the southern part of the Wei River was better than that north of the Wei River. Areas containing high nitrate concentrations were mainly located in the central and western parts of the Guanzhong Basin. The percentages of low risk (<45 mg/L), high risk (45–100 mg/L), and very high risk (>100 mg/L) of nitrate pollution in the study area were 90.58%, 8.9%, and 0.52%, respectively. The HHR assessment results indicated that people in the 6–12 month age group were more likely to suffer from health complications due to a higher nitrate concentration, followed by 6–11 years, 21–65 years, 18–21 years, ≥65 years, 11–16 years, and 16–18 years age groups, which was mainly due to the different exposure parameters. The results of this study will be useful in regional groundwater management and protection.

## 1. Introduction

Groundwater is extensively used for agricultural, industrial, and drinking purposes in many arid and semi-arid regions (e.g., Western United States, Algeria, Egypt, China, and Australia) where rainfall is infrequent and surface water is scarce [1,2,3,4]. Groundwater accounts for only 29.9% of all global freshwater resources [1,5], and water resource shortages have become one of the most important challenges to humankind [4,6]. Moreover, groundwater resources have drastically declined not only in quality but also in quantity due to untreated effluents from industrial and agricultural development, expanding urbanization, population growth, inadequate sanitation, and pollutant runoff in arid and semi-arid regions [2,7,8]. Hydrochemical characteristics are widely used to indicate the source of the main components of ions, types of groundwater, water–rock interactions, and groundwater reservoir environments [9]. Knowledge of hydrochemical characteristics is useful for evaluating groundwater quality because it provides an understanding of groundwater suitability for various purposes. Investigations have shown that exposure to potentially toxic chemicals, such as heavy metals, fluorides, and nitrate in groundwater can pose great risks to human health [10,11]. 

Nitrate is one of the main groundwater pollutants and significantly impacts groundwater quality. Groundwater pollution can affect human health and is the most widespread source of health problems in arid and semi-arid regions around the world [12,13,14]. Excessive nitrate in drinking groundwater has been reported to cause various health complications, including abortions, blue baby syndrome, increased risk of methemoglobinemia and gastric cancer, damage to stomach lining, mouth ulceration, and reproductive damage [15,16,17]. Therefore, the concentration of nitrate in groundwater should be monitored. Adimalla et al. [12] evaluated the fluoride and nitrate pollution of groundwater in the semi-arid region of Nirmal Province, South India and found that high fluoride and nitrate concentrations in water caused health issues in humans. Adimalla and Li [18] suggested that high fluoride and nitrate concentrations in water in the semi-arid region, Telangana State, India posed a risk to residents and made water unsuitable for drinking. Wu et al. [17] studied severe nitrate pollution and health risks of a coastal aquifer, and the results showed that groundwater nitrate concentrations exerted non-carcinogenic health risks for different age groups. Nejatijahromi et al. [19] analyzed the stable isotope ratios (δ^15^N-NO_3_^−^, δ^18^O-NO_3_^−^, δ^18^O-H_2_O, and δD-H_2_O) of groundwater samples and inferred that the main nitrate sources in groundwater were chemical fertilizers and treated wastewater near Tehran, Iran. Similar studies in other regions have also been carried out [15,20,21].

The Guanzhong Basin is located on the “Silk Road Economic Belt” in a semi-arid area in Shaanxi Province, China, and has supported significant agriculture since the Qin Dynasty [9,22]. Moreover, this region is the political, economic, and cultural center of Shaanxi Province and has a significant influence on China’s national regional economic pattern [23,24]. Sixty percent of the population, 80% of industry, and 52% of cultivated land in Shaanxi are located in the Guanzhong Basin. However, natural factors and human activities have caused many environmental problems, such as water resource shortages, which have exacerbated water pollution [25,26]. Large amounts of irrigation water and fertilizer are used each year in traditional farming areas, causing nitrate to pollute groundwater which is mainly used for human consumption. Polluted groundwater may cause great risks to human health [13,27], making it important to study the current status of nitrate pollution and associated health risks in the Guanzhong Basin.

Therefore, the main objective of this study was to (1) study the hydrogeochemical characteristics, hydrochemical facies of groundwater, and their formation mechanisms; (2) appraise the overall groundwater quality for drinking purposes using the water quality index (WQI); (3) analyze the distribution and main sources of nitrate in the Guanzhong Basin; and (4) assess the non-carcinogenic health risks posed by drinking nitrate-contaminated groundwater for different age groups (6–12 months, 6–11 years, 11–16 years, 16–18 years, 18–21 years, 21–65 years, and ≥65 years). The outcome of this study provides essential information for policy and decision makers to take appropriate action to improve the groundwater quality in the study region.

## 2. Study Area

The Guanzhong Basin covers an area of 2 × 10^4^ km^2^ and is located at approximately 33°00’–35°20′ N and 106°30′–110°30′ E in the middle of Shaanxi Province, China [28,29]. Its average elevation is about 400 m above sea level (a.s.l.) [28]. The Guanzhong Basin is also one of the largest basins of the Yellow River Catchment in China [30] and is located between the Qinling Mountains in the south with a maximum elevation of 3767 m a.s.l. and the North Mountains in the north with a maximum elevation of 1678 m a.s.l. [31], as shown in Figure 1a. The North Mountains are covered by quaternary loess and loess-like deposits, which are the main source of sediments in the rivers [26]. The population of the Guanzhong Basin includes the population of Xi’an (8.07 million), Weinan (5.47 million), Xianyang (4.98 million), and Baoji (3.76 million), accounting for about 60% of the total population (37.2 million) of Shaanxi Province. Moreover, the Guanzhong Basin is an important section of the Silk Road economic belt and has been part of the most important political, economic, cultural, social, and educational areas of the world [9,32,33]. 

This region is characterized by a temperate continental monsoon climate with an annual average rainfall of 573 mm, approximately 78% of which occurs from May to October [9,26]. Evaporation generally ranges between 1000 and 1200 mm per year [34]. The annual average temperature is 13.3 °C, the average annual daylight is 1500–2200 h, and the frost-free period is 200–220 days. The Wei River, the largest tributary (502.4 km) of the Yellow River, receives 538 million m^3^ of the average annual runoff in Shaanxi Province [28].

The Guanzhong Basin is a Cenozoic fault basin, and subsidence in the basin began in the Late Eocene and continued throughout the Miocene and Pliocene epochs, leading to the deposition of a thick sequence of tertiary fluvial–lacustrine clastic rocks [34]. Therefore, the main water-bearing rock formation of the Guanzhong Basin is loose sediment-type pore water-bearing formation (L) formed in the quaternary, as shown in Figure 1b. The aquifers also include carbonate fissure-karst water-bearing rock formations (CA) in the North Mountains and magmatic fissure-karst water-bearing rock formations (MA) and metamorphic fissure water-bearing rock formations (ME) in the Qinling Mountains [26]. Groundwater can be recharged by surface water due to the loose sediment pore water-bearing formations, which have good permeability and high leakage coefficients.

## 3. Materials and Methods

### 3.1. Sample Collection and Analysis

In the study region, 191 groundwater samples were collected from a phreatic aquifer in the central-western Guanzhong Basin in 1999, as shown in Figure 1. All samples were analyzed for various hydrochemical parameters, such as pH, total dissolved solids (TDS), electrical conductivity (EC), total hardness (TH, as CaCO_3_), potassium and sodium (K^+^+Na^+^), calcium (Ca^2+^), magnesium (Mg^2+^), chloride (Cl^−^), sulfate (SO_4_^2−^), bicarbonate (HCO_3_^−^), and nitrate (NO_3_^−^). pH and TDS were measured immediately in the field using portable devices, and EC was calculated from TDS. The TH, Ca^2+^, and Mg^2+^ were analyzed by a titrimetric method using EDTA. K^+^ + Na^+^ was measured by flame atomic absorption spectrophotometry. SO_4_^2−^, Cl^−^, and NO_3_^−^ were determined by ion chromatography, and HCO_3_^−^ was determined by alkalinity titration. 

For each groundwater sample, the computed ionic-balance-error (IBE) was observed to be within the acceptable limit of ±5% and was calculated by Equation (1) to ensure the accuracy of the analysis [9,35]:(1)$$\mathrm{IBE}=\frac{\sum Cations-\sum Anions}{\sum Cations+\sum Anions}\times 100.$$

All cations and anions were expressed in meq/L. The calculated results showed that the IBE varied from −4.43 to 4.04.

### 3.2. Water Quality Index (WQI)

The water quality index (WQI) reduces water parameters to a single number to assess the overall water quality at a certain location, and was computed to evaluate whether groundwater samples were suitable for drinking [36,37]. WQI was calculated using the 12 measured parameters at each site, in three steps in the current study. The first step was to estimate the relative weight (*W_i_*) of each parameter, as given in Equation (2):(2)$$W_{i}=\frac{w_{i}}{\sum_{i=1}^{n}w_{i}}\;\;\;$$
where *w_i_* is the weight of each parameter, and *n* is the number of hydrochemical parameters. The value of *w_i_* ranged from 1 to 5 according to the impact of the contaminant on human health [37]. The second step was to estimate the quality rating scale (*Q_i_*) of each parameter using Equation (3):(3)$$Q_{i}=\frac{C_{i}-C_{ip}}{S_{i}-C_{ip}}\times 100\;$$
where *C_i_* is the concentration of each parameter in mg/L, *S_i_* is the standard permissible value for the *i* parameter in mg/L, and *C_ip_* is the ideal value of the *i* parameter in pure water (consider $C_{jp}$ = 0 for all, except the pH value where $C_{jp}$ = 7). The third step was to calculate the WQI using Equation (4):(4)$$WQI=\overset{n}{\underset{i=1}{\sum }}W_{i}\times Q_{i}.\;\;$$

The calculated values of *w_i_*, *W_i_*, and *S_i_* are shown in Table 1.

### 3.3. Human Health Risk (HHR) Assessment via Water Pathway

The HHR assessment, established by the United States Environmental Protection Agency (USEPA), is a widely used tool for evaluating human health risk [11,12,18,35]. The human body primarily absorbs contaminants via drinking water [14,38], and so only the health effects of contaminants introduced through drinking water were considered in the study region. Nitrate concentration was selected for HHR assessment in the current study. The chronic daily intake (*CDI*), hazard quotient (*HQ*), and the hazard index inducing from nitrate (*HI_nitrate_*) were calculated by Equations (5) and (6), respectively. The detailed reference values of each parameter used for the calculation are presented in Table 2.
(5)$$CDI=\frac{C_{w}\times IR\times EF\times ED}{BW\times AT}\;\;$$
(6)$$HI_{nitrate}=HQ=\frac{CDI}{RfD}\;$$

## 4. Results and Discussion

### 4.1. Groundwater Chemistry

#### 4.1.1. Physiochemical Parameters

Table 3 lists descriptive statistical parameters (min, max, and mean) for the analyzed groundwater samples of the study region. Table 3 also includes details of drinking water quality limits (*S_i_*), and the number and percentage of samples which exceeded the permissible limit in the absence of an alternate source (PLAAS). In this region, the pH of groundwater samples varied from 6.9 to 8.4 with a mean of 7.68. Ninety-eight percent of samples were alkaline, and all were within the recommended limits of 6.5–8.5 [39]. The TDS concentrations ranged from 110.7 to 2978.73 mg/L with a mean of 516.81 mg/L, and fourteen samples (7.33% of total samples) exceed the recommended limit of 1000 mg/L for drinking water, as shown in Table 3. The TH concentration ranged from 1.96 to 530.5 mg/L with an average of 64.87 mg/L, and 98.95% were within the standard limit (450 mg/L), indicating a suitable quality for drinking. EC is used to measure the ability of water to conduct an electric current, and an elevated EC reveals the enrichment of silt in water [12,40]. The EC of groundwater samples ranged from 201.26 to 4445.86 μS/cm with a mean of 885.75 μS/cm, as shown in Table 3. Also, 90.05% of samples were below 1500 μS/cm, indicating a low enrichment of salts in this area.

For cation chemistry, K^+^+Na^+^ (mean = 102.12 mg/L) were the dominant ions in the groundwater of the study region, followed by Ca^2+^ (mean = 49.14 mg/L) and Mg^2+^ (mean = 31.91 mg/L). The concentration of K^+^+Na^+^ ranged from 2.07 to 400.89 mg/L, and 17.80% of groundwater samples exceeded the acceptable limit of 200 mg/L for drinking. Cation exchange and frequent evaporation may be the main causes of the high Na^+^ concentration in groundwater in the study region [11,41]. Ca^2+^ and Mg^2+^ are also key hydrochemical elements for human health in groundwater [42]. As shown in Table 3, Ca^2+^ differed from 7.21 to 124.2 mg/L, indicating that all groundwater samples were under the standard limit of 200 mg/L [39]. The Mg^2+^ concentration ranged from 1.2 to 293.57 mg/L, and three samples exceeded the recommended limit of 200 mg/L for drinking.

The average concentration of anions in all samples followed the order HCO_3_^−^ > SO_4_^2−^ > Cl^−^ > NO_3_^−^. The concentration of HCO_3_^−^ varied from 109.83 to 1020.22 mg/L with a mean of 374.79 mg/L. Mineral dissolution, which controls the groundwater chemistry, is the main reason for the high HCO_3_^−^ concentration [43]. In the present study, the concentration of SO_4_^2−^ ranged from 0 to 634.97 mg/L, with an average of 70.01 mg/L. The Cl^−^ content varied from 2.48 to 1556.39 mg/L with a mean of 57.09 mg/L. Overall, 6.28% and 5.76% of groundwater samples were within the acceptable limit of 250 mg/L for SO_4_^2−^ and Cl^−^ in the study region, respectively. The NO_3_^−^ content varied between 0 and 90 mg/L with a mean of 18.26 mg/L. Forty-seven groundwater samples exceeded the standard limit of 20 mg/L, indicating that 24.61% of groundwater samples in the study region were unsuitable for drinking, and nitrogen pollution was the most serious form of pollution.

#### 4.1.2. Hydrochemical Types of Groundwater

Hydrochemical classifications of groundwater are governed by major ions. A Durov diagram, as shown in Figure 2, is a useful graphical tool that is widely used to identify the chemical relationship and evolution of groundwater samples [38,44]. As shown in Figure 2, the dominant hydrochemical facies of the groundwater samples were HCO_3_-Na and HCO_3_-Ca. For cations, the majority of samples were also observed in B (no dominant), indicating that the cations were also predominated by the mixed type. Figure 2 reveals that more than 90% of groundwater samples had a TDS less than 1000 mg/L in which HCO_3_^−^ was the main anion. Approximately 10% of groundwater samples were brackish water with TDS values ranging from 1000 to 3000 mg/L. High nitrate concentrations [27] in this field were characterized by fresh water (TDS < 1000 mg/L) [44], especially TDS values < 500 mg/L.

#### 4.1.3. Mechanisms of Hydrogeochemistry

Interactions between groundwater and aquifers can significantly affect water chemistry [45]. To determine the sources of major ions, bivariate diagrams were constructed for the groundwater samples in the study region, as shown in Figure 3. Generally, if halite dissolution was the only source of Na^+^ and Cl^−^, the data points of groundwater samples were distributed along a 1:1 line. As shown in Figure 3a, the plot of Na^+^ vs. Cl^−^ shows a clear Na^+^ enrichment in the groundwater samples, possibly due to halite dissolution, weathering of silicate minerals, and ion exchange [9,13,46].

Gypsum weathering is a major process which contributes to the presence of Ca^2+^ in groundwater, and is the only source of Ca^2+^ and SO_4_^2−^ when the ratio of the two ions is approximately 1 [13]. As shown in the bivariate diagrams of Ca^2+^ and SO_4_^2−^ in Figure 3b, most samples were plotted below the 1:1 line, indicating that other sources contributed to the presence of Ca^2+^, such as the dissolution of calcite and dolomite [9,47]. In theory, if calcite and dolomite dissolution were the only sources of Ca^2+^ and Mg^2+^, the HCO_3_^−^/(Ca^2+^+Mg^2+^) ratio should lie from 1:1 to 2:1 depending on the amount of CO_2_ [46,47]. The specific reactions involved in this process are as follows:(7)$$CaCO_{3}+H^{+}↔HCO_{3}{}^{-}+Ca^{2+}$$
(8)$$CaCO_{3}+H_{2}O+CO_{2}↔2HCO_{3}{}^{-}+Ca^{2+}\;$$
(9)$$CaMg{\left(CO_{3}\right)}_{2}+2H^{+}↔2HCO_{3}{}^{-}+Ca^{2+}+Mg^{2+}\;$$
(10)$$CaMg{\left(CO_{3}\right)}_{2}+2H_{2}O+CO_{2}↔4HCO_{3}{}^{-}+Ca^{2+}Mg^{2+}$$

As shown in Figure 3c, HCO_3_^−^/(Ca^2+^+Mg^2+^) in most groundwater samples ranged from 1:1 to 2:1, indicating that the primary source of Ca^2+^ and Mg^2+^ was dolomite dissolution. However, more than half of the samples were plotted below the 2:1 line, as shown in Figure 3d, suggesting that the calcite dissolution had a minor effect on water chemistry. According to this result, the ratio of Na^+^/Cl^−^ exceeded 1, indicating that ion exchange between Ca^2+^ and Na^+^ may have occurred and was responsible for the excess HCO_3_^−^. This ion exchange can be expressed as follows:(11)$$Ca^{2+}+2NaX↔CaX_{2}+2Na^{+}.\;$$

Ion exchange was also studied using the chloro-alkaline indices (CAI-I and CAI-II) proposed by [48]. CAI-I and CAI-II were calculated using Equations (12) and (13) to interpret the ion exchange behavior between groundwater and the prevailing environment [10]. All ions were expressed in meq/L.
(12)$$CAI-I=\frac{Cl^{-}-\left(Na^{+}+K^{+}\right)}{Cl^{-}}\;\;$$
(13)$$CAI-II=\frac{Cl^{-}-\left(Na^{+}+K^{+}\right)}{SO_{4}{}^{2-}+HCO_{3}{}^{-}+CO_{3}{}^{2-}+NO_{3}{}^{-}}$$

If CAI-I and CAI-II are negative, ion exchange occurred between Ca^2+^ or Mg^2+^ in the groundwater and Na^+^ and K^+^ in the rocks or soil, and a reverse ion exchange occurred when positive values of CAI-I and CAI-II were obtained [4,10]. The CAI-I and CAI-II values obtained in this study are shown in Figure 4. The results revealed that 92.67% of groundwater samples possessed negative CAI-I and CAI-II values, indicating that ion exchange may have occurred between Ca^2+^ or Mg^2+^ in the groundwater and Na^+^ and K^+^ in the rocks or soil. 

### 4.2. Groundwater Quality for Drinking

The water quality index (WQI) was used to assess the groundwater quality for drinking in this study. Groundwater quality can be classified into five categories [13] based on the WQI value: excellent (WQI < 25), good (25 < WQI < 50), poor (50 < WQI < 100), very poor (100 < WQI < 150), and unsuitable for drinking (WQI > 150). As shown in Figure 5, the groundwater quality ranged from excellent to extremely poor. The groundwater in the Guanzhong Basin primarily contains good and medium-quality water. The good quality water was mainly located south of the Wei River, while the medium water was mainly distributed north of the Wei River. Small amounts of poor and extremely poor water were scattered throughout the area northwest of Xianyang City, and some excellent water was scattered south of the Wei River. It can be seen that the groundwater south of the Wei River was better than that north of the Wei River, which was mainly related to the topography and agriculture of the Guanzhong Basin. Specifically, there is a lower population and less agriculture south of the Wei River than north of the Wei River, primarily due to the presence of the Qinling Mountains. Additionally, the surface water system is more developed and groundwater usage is lower in the southern mountainous areas, so the probability of usage of polluted groundwater is relatively lower.

### 4.3. Distribution and Occurrence of Nitrate

The groundwater nitrate concentration maps in Figure 6 show that high nitrate concentrations are distributed mainly in the area north of the Wei River, likely due to the higher population in this region. The nitrate contamination may originate from many human activities, such as the extensive use of fertilizers, manures, and insecticides in agriculture, septic tank leakage, and organic matter effluents [12]. The use of agricultural fertilizers was the principal cause for higher NO_3_^−^ concentration in the groundwater of the study region. In addition, among the 191 groundwater samples, 47 samples had NO_3_^−^ concentrations higher than the standard limit of 20 mg/L, as shown in Table 3. The low nitrate concentration in groundwater in the area south of Xi’an is that the main drinking water in Xi’an is surface water, which has a lower impact on groundwater. Additionally, industrial ammonia nitrogen also contributed to the nitrate concentration in Guanzhong. Wan [49] studied the temporal and spatial variation of mineral nitrogen pollution in groundwater in the Guanzhong Basin and found that variations in industrial ammonia nitrogen emissions were consistent with changes in the nitrate content in groundwater. It can be seen that agricultural activities had a greater effect than industry on the nitrate content in groundwater in the Guanzhong Basin. Moreover, according to the levels of nitrate risk defined in [27], the percentages of low risk (<45 mg/L), high risk (45–100 mg/L), and very high risk (>100 mg/L) were 90.58%, 8.9%, and 0.52%, respectively. The concentration of nitrate in the study area poses a great risk to human drinking water. Some studies [14,50,51] have indicated that long-term drinking of high-nitrate groundwater can cause methemoglobinemia in humans. 

### 4.4. Evaluation of Human Health Risk (HHR)

The total hazard indices (*HI*_total_) associated with nitrate concentration via ingestion were estimated for different age groups (6–12 months, 6–11 years, 11–16 years, 16–18 years, 18–21 years, 21–65 years, and ≥65 years). The statistical results of non-carcinogenic health risks are shown in Table 4. The *HI*_total_ values ranged from 0.006868 to 13.0495 with a mean of 1.129466 for 6–12 months. The *HI*_total_ values for age 6–11 years varied from 0.002816 to 5.34983 with an average of 0.463043. The values for age 11–16 years, 16–18 years, 18–21 years, 21–65 years, and ≥65 years ranged from 0.002099 to 3.98755, 0.001646 to 3.12685, 0.002163 to 4.11058, 0.002332 to 4.43052, and 0.002133 to 4.05234, respectively. The obtained *HI_total_* values showed that 69 groundwater samples exceeded the acceptable limit *HI* = 1 for the 6–12 months group, while 26, 18, and 8 groundwater samples exceeded the acceptable limit for 6–11 years, 11–16 years, and 16–18 years age groups, respectively. In addition, for 18–21 years, 21–65 years, and ≥65 years, 18, 22, and 18 samples exceeded the acceptable limit, and their means were 0.355782, 0.383474, and 0.350742, respectively. The HHR results indicated that the 6–12 month age group was more likely to suffer from health complications due to a higher nitrate concentration, followed by 6–11 years, 21–65 years, 18–21 years, ≥65 years, 11–16 years, and 16–18 years age groups based on the mean values. This order is mainly related to the discrepancy of exposure parameters [27]. Figure 7 was plotted to show details of the results of non-carcinogenic health risk for different age groups.

Moreover, a percentage bar graph of the hazard indices of the different age groups exceeding the acceptable limit of HI = 1 is shown in Figure 8. As shown in Figure 8, the same nitrate content in groundwater had different effects on the non-carcinogenic risks in different age groups, which ranged from high to low as: 6–12 months > 6–11 years > 21–65 years > 18–21 years > ≥65 years > 11–16 years > 16–18 years. This may be related to the body’s immunity and metabolism of different age groups. These results are especially useful for decision makers and will provide them with general risk information to help make proper decisions. Safe drinking water should be supplied for local residents and groundwater management should be further implemented in the study region.

### 4.5. Sustainable Groundwater Quality Management

Due to the importance of groundwater for maintaining the life of humans and plants, the sustainable development and management of groundwater resources is critical, especially in poor water quality zones. The results of groundwater quality and human health risk assessments have indicated that groundwater in the study region is not totally suitable for human consumption. Therefore, some management strategies are recommended to improve the groundwater quality, reduce the human health risk, and enhance the sustainable groundwater quality management in the study area [35].
The use of large-scale fertilizers has resulted in nitrogen pollution in the study area, so reducing the use of fertilizers, such as using organic fertilizers, is crucial for reducing nitrogen pollution.Since industrial ammonia nitrogen emissions are one of the main sources of nitrogen pollution in the study area, effective treatment of industrial ammonia nitrogen after discharge is necessary, such as industrial nitrogen fixation.Additionally, strengthening scientific research, improving the monitoring network, and enhancing cooperation and data sharing are important and necessary for ensuring the sustainable development of groundwater [52].In this paper, regrettably, surface water samples were not obtained, so there is no comparative analysis of the relationship and changes between surface water quality and groundwater quality, and it is hoped that supplementary research can be carried out in the future.

## 5. Conclusions

In this study, various physiochemical parameters of 191 groundwater samples were analyzed to understand the geochemistry of groundwater and nitrate contamination and the health risks it poses to inhabitants of the Guanzhong Basin, China. The outcomes of the study can be summarized as follows:HCO_3_-Na and HCO_3_-Ca species were the dominant hydrochemical facies in the groundwater samples in the study region. Bivariate diagrams and chloro-alkaline indices were used to study the hydrogeochemistry mechanisms. The results showed that halite dissolution, the weathering of silicate minerals, and ion exchange were the major sources responsible for Na^+^ abundance. Gypsum weathering and dolomite dissolution were the major processes controlling the Ca^2+^ concentration, while dolomite dissolution was the primary source of Mg^2+^. Moreover, ion exchange between Ca^2+^ or Mg^2+^ in the groundwater with Na^+^ and K^+^ in the rocks or soil may have occurred in the groundwater of the study region.The groundwater in the Guanzhong Basin predominantly contains good and medium-quality water. The good water was mainly distributed in the area south of the Wei River, while the medium-quality water was mainly distributed in the area north of the Wei River. Very little poor water or extremely poor water was found in the northwest of Xianyang City, and some excellent water was scattered throughout the area south of the Wei River. Generally, the groundwater in the area south of the Wei River was better than that in the area north of the Wei River.High nitrate areas were mainly distributed in the area north of the Wei River. The nitrate contamination potentially originated from a large number of human activities, such as the extensive use of fertilizers, manures, and insecticide in agriculture, septic tank leakage, and organic matter effluent. Moreover, industrial ammonia nitrogen also played a role in the pollution of nitrate contamination in Guanzhong. Additionally, the percentages of low risk (<45 mg/L), high risk (45–100 mg/L), and very high risk for nitrate pollution (>100 mg/L) were 90.58%, 8.9%, and 0.52%, respectively.The HHR results indicated that the 6–12 month age group was more likely to suffer from health complications due to a higher nitrate concentration, followed by 6–11 years, 21–65 years, 18–21 years, ≥65 years, 11–16 years, and 16–18 years age groups based on the average values. These results are especially useful for decision makers and will provide them with general risk information to allow them to make appropriate decisions.

## Figures and Tables

**Figure 1 ijerph-16-04246-f001:**
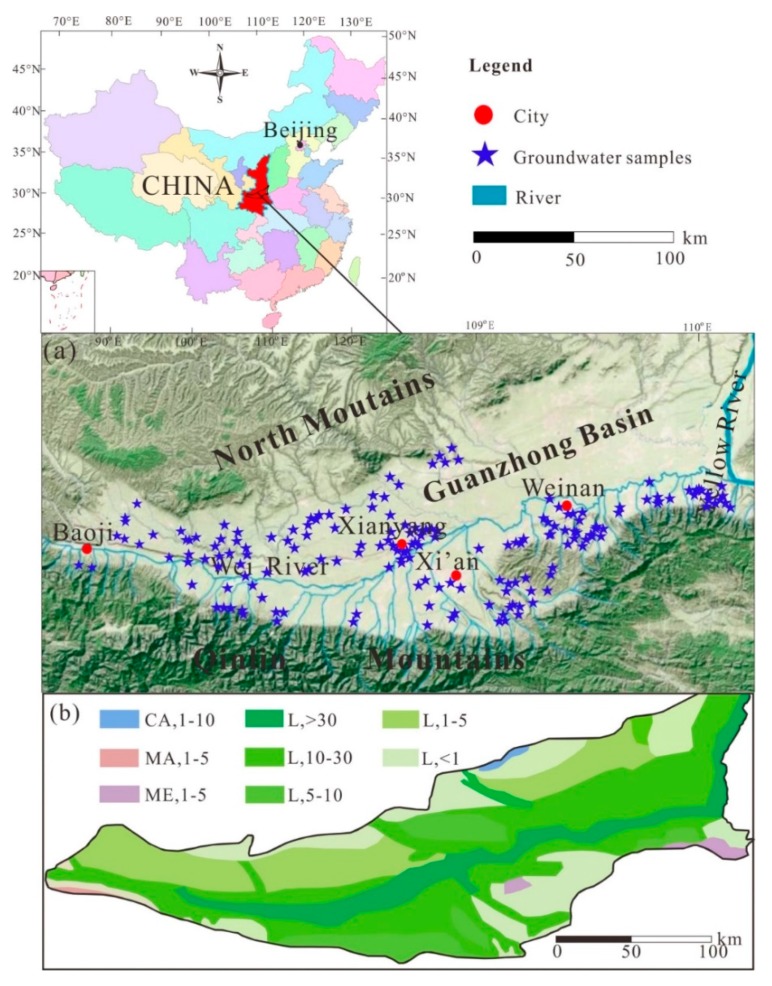
Study area showing (**a**) geographical locations of sampling sites and (**b**) the geological map. Fissure-karst water-bearing formations: CA, carbonate; MA, magmatic; ME, metamorphic; L, loose porous sediment. The number after the letter represents the mass specific storage (t/h*m).

**Figure 2 ijerph-16-04246-f002:**
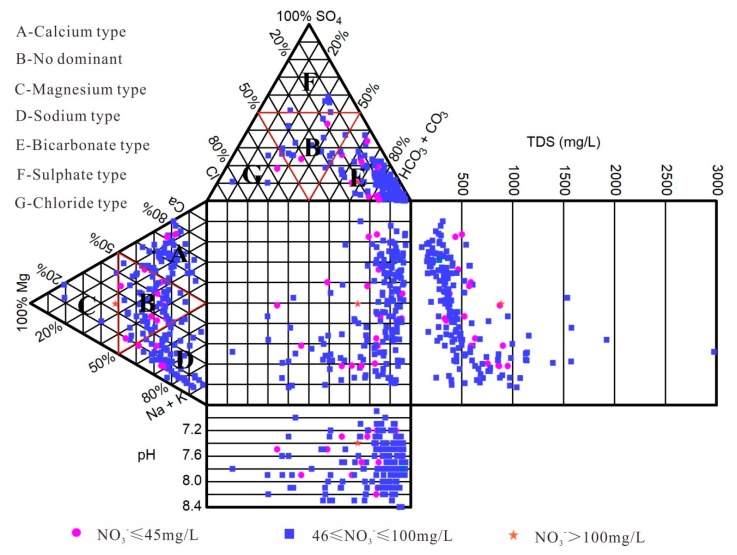
Durov diagram representing groundwater types.

**Figure 3 ijerph-16-04246-f003:**
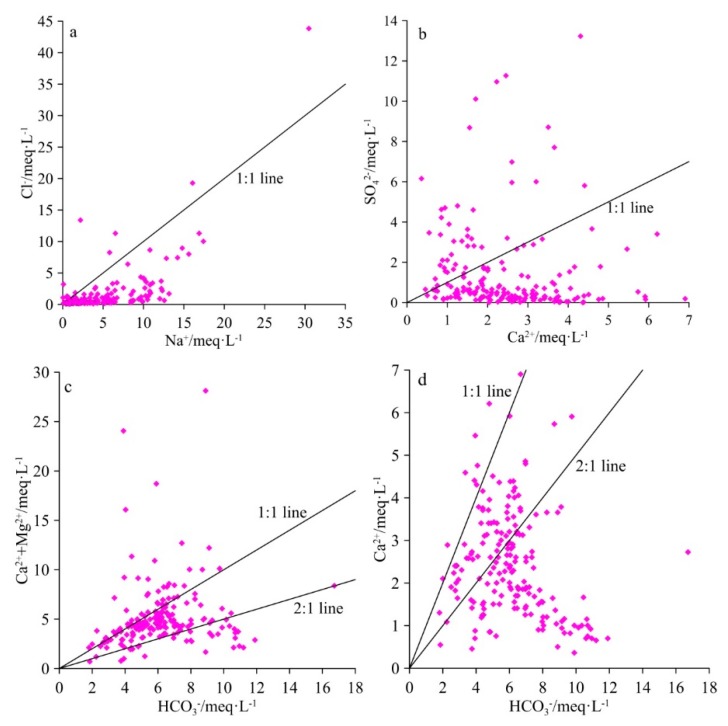
Bivariate diagrams of ionic concentrations in groundwater.

**Figure 4 ijerph-16-04246-f004:**
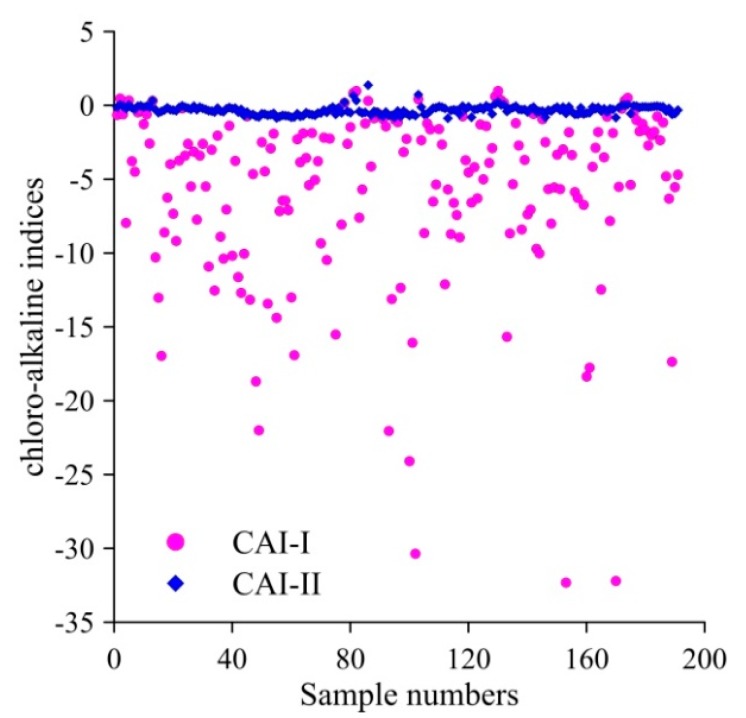
Scatter diagram of the chloro-alkaline indices (CAI-I and CAI-II) of groundwater samples in the study region.

**Figure 5 ijerph-16-04246-f005:**
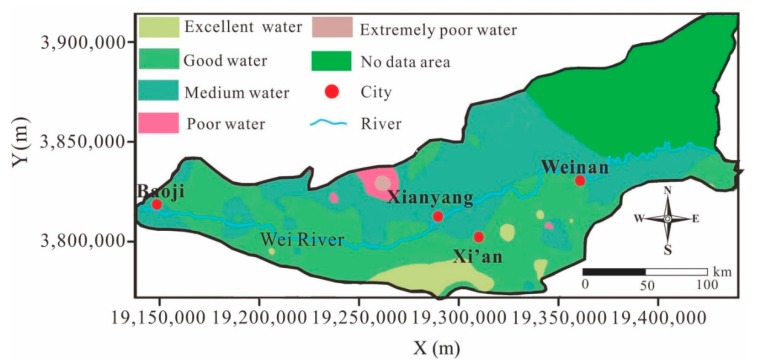
Distribution of the water quality index (WQI) in the groundwater of the study region.

**Figure 6 ijerph-16-04246-f006:**
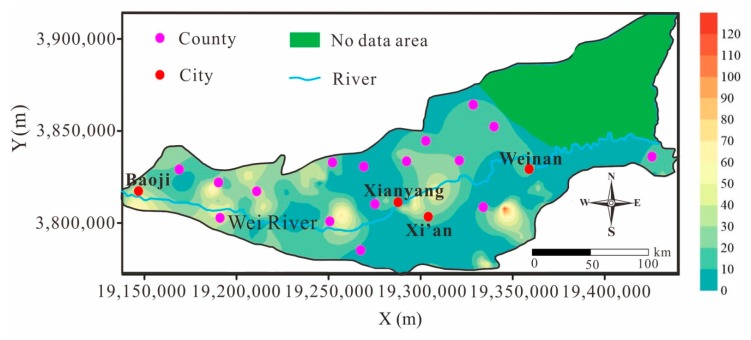
Spatial distribution map of NO_3_^−^ in the Guanzhong Basin, China.

**Figure 7 ijerph-16-04246-f007:**
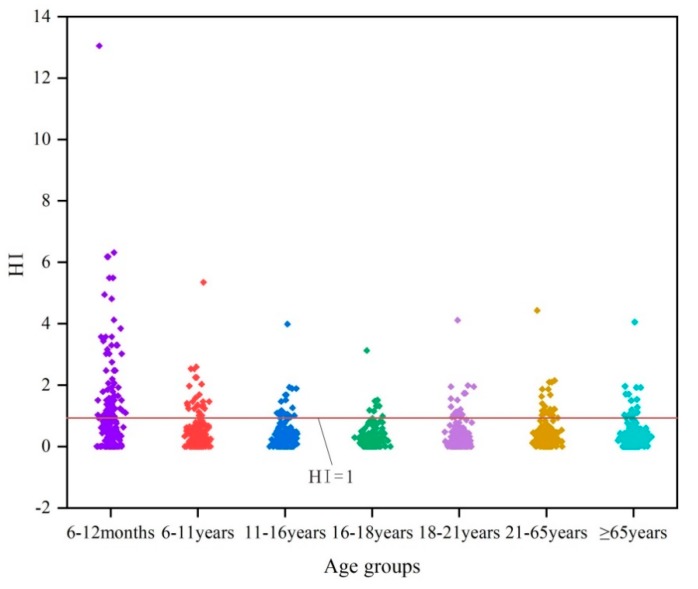
The results of non-carcinogenic health risk for different age groups.

**Figure 8 ijerph-16-04246-f008:**
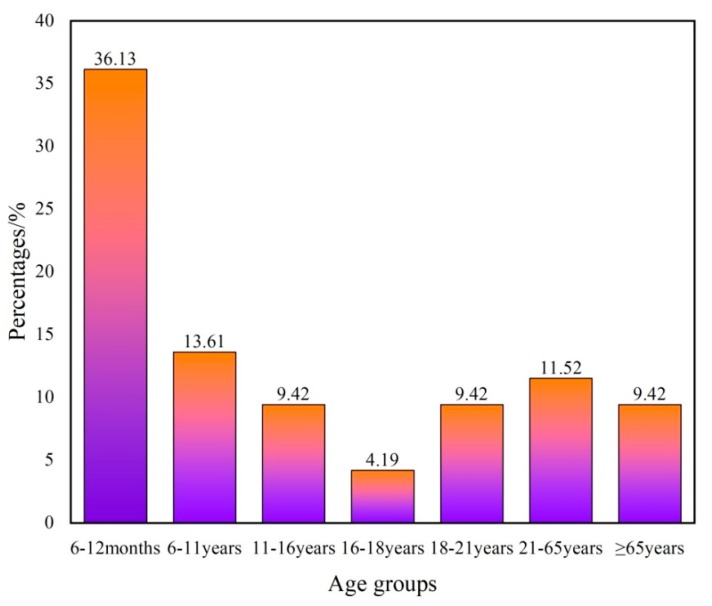
Percentage bar graph of the hazard indices of the different age groups exceeding the acceptable limit of HI = 1.

**Table 1 ijerph-16-04246-t001:** The values of *w_i_*, *W_i_*, and *S_i_* of different parameters. TDS, total dissolved solids; TH, total hardness; EC, electrical conductivity.

Parameters	k^+^+Na^++^	Ca^2+^	Mg^2++^	Cl^−^	SO_4_^2−^	HCO_3_^−^	NO_3_^−^	pH	TDS	TH	EC
*w_i_*	2	3	3	4	3	3	5	3	5	3	4
*W_i_*	0.053	0.079	0.079	0.105	0.079	0.079	0.132	0.079	0.132	0.079	0.105
*S_i_*	200	200	150	250	250	500	20	8.5	1000	450	1500

**Table 2 ijerph-16-04246-t002:** Key parameters for computing the exposure risk of nitrate through drinking water.

Parameters	Units	6–12 Months	6–11 Years	11–16 Years	16–18 Years	18–21 Years	21–65 Years	≥65 Years
Ingestion rate (IR)	L/day	1	1.32	1.82	1.78	2.34	2.94	2.73
Exposure frequency (EF)	day/year	365	365	365	365	365	365	365
Exposure duration (ED)	year	6	6	6	6	30	30	30
Body weight (BW)	kg	9.1	29.3	54.2	67.6	67.6	78.8	80
Average time (AT)	days	2190	2190	2190	2190	10,950	10,950	10,950
Concentration of element (*C_w_*)	mg/L	Present study						
Reference dose of nitrate (RfD)	-	1.6						

**Table 3 ijerph-16-04246-t003:** Statistical analysis of physiochemical parameters.

Parameters	Units	Min.	Max.	Mean	*S_i_*	Number of Samples Exceeding the PLAAS	% of Samples Exceeding the PLAAS
pH	-	6.9	8.4	7.68	6.5–8.5	0	0.00
TDS	mg/L	110.7	2978.73	516.81	1000	14	7.33
TH	mg/L	1.96	530.5	64.87	450	2	1.05
EC	μS/cm	201.26	4445.86	885.75	1500	19	9.95
k^+^+Na^+^	mg/L	2.07	400.89	102.12	200	34	17.80
Ca^2+^	mg/L	7.21	124.2	49.14	200	0	0.00
Mg^2+^	mg/L	1.2	293.57	31.91	150	3	1.57
Cl^−^	mg/L	2.48	1556.39	57.09	250	11	5.76
SO_4_^2−^	mg/L	0	634.97	70.01	250	12	6.28
HCO_3_^−^	mg/L	109.83	1020.22	374.79	500	31	16.23
NO_3_^−^	mg/L	0	90	18.26	20	47	24.61

PLAAS: permissible limit in the absence of an alternate source.

**Table 4 ijerph-16-04246-t004:** The statistical results of non-carcinogenic health risks in the study region.

HI_total_	Max.	Min.	Mean	Num.	Per.
6–12 months	13.0495	0.006868	1.129466	69	36.13%
6–11 years	5.34983	0.002816	0.463043	26	13.61%
11–16 years	3.98755	0.002099	0.345133	18	9.42%
16–18 years	3.12685	0.001646	0.270637	8	4.19%
18–21 years	4.11058	0.002163	0.355782	18	9.42%
21–65 years	4.43052	0.002332	0.383474	22	11.52%
≥65 years	4.05234	0.002133	0.350742	18	9.42%

Num.: the number of groundwater samples exceeding the acceptable limit of HI = 1; Per.: a percentage of the number of water samples exceeding the acceptable limit of HI = 1 relative to the number of all groundwater samples. HI, hazard index.

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
