# Peer review of "Assessment of Groundwater Quality and Human Health Risk (HHR) Evaluation of Nitrate in the Central-Western Guanzhong Basin, China"

_ijerph, 2019, doi:10.3390/ijerph16214246_

Round 1

Reviewer 1 Report

The study of Zhang et al. tackles an important problem of the water quality in the Central-Western Guanzhong Basin (China) in terms of urbanization and intensification of agriculture. The authors tried to evaluate groundwater chemistry for drinking purposes using WQI and to identify a distribution of nitrate concentrations. The concentrations of hydrochemical and physical parameters were monitored presumably once (this is not explained). Results are validated against spatial and statistical analysis. The manuscript brings important qualitative data that can be useful for water management. The paper is well-organized, I also did not find any errors in the presentation of results and discussions. Also worthy of note is the comprehensive approach to the study. I have only some minor remarks, addressed to the methods and study area, which should be considered (discussed) in a final version. This will help to improve the manuscript and make clear a finding beyond the previous researches.

Study area

To understand the local situation, we also need to know the local demography. What is the population (density of population) in this region? In the introduction, the authors raise the problem of water quantity. If you have data about the water level changes, please add them to the text. Evaporation generally ranges between 1000 and 1200 mm per year [9]

Please verify the reference. In Xu et al. (2019) this data is cited from Dou (2018).

The lack of historical groundwater chemistry data is notably evident. Are there any previous data?

Methods

As mentioned above: There is even no information about the date of sampling. 1) When were the groundwater samples (191) collected? 2) How many replicates were made? Was the concentrations of hydrochemical and physical parameters were monitored in the same year/period? What methods were used to develop Figures 5 and 6. Which classification (interpolation) technique was used?

In addition, it is worth showing a scale bar on both figures (instead of a north arrow). Figure 6 is illegible, please reorganize map composition. Legend entry “County” is not evident - can you present counties with point feature type?

Author Response

Dear reviewer:

Thank you so much for giving us warning and suggestion to revise our manuscript. We have considered comments from you carefully and have made revisions to address your concerns. It is worth noting that all revised details are marked in green.

Sincerely,

Hui Qian

Point 1: To understand the local situation, we also need to know the local demography. What is the population (density of population) in this region?

Response 1: Thanks for the reviewer’s kind suggestion. The local demography has been introduced in detail. The revised details can be found in the part of “Study Area”.

Point 2: In the introduction, the authors raise the problem of water quantity. If you have data about the water level changes, please add them to the text.

Response 2: Thanks for the reviewer’s kind suggestion. Unfortunately, we do not have the data available. I am very sorry about this.

Point 3:  Evaporation generally ranges between 1000 and 1200 mm per year [9]. Please verify the reference. In Xu et al. (2019) this data is cited from Dou (2018).

Response 3: Thanks for the reviewer’s kind warning. After verification, there is a problem in reference 9. So, the reference 9 has been replaced by reference 34. The revised details can be found in section of “Study Area”.

Point 4:  The lack of historical groundwater chemistry data is notably evident. Are there any previous data?

Response 4: Thanks for the reviewer’s kind suggestion. Unfortunately, we do not have the data available. I am very sorry about this.

Point 5: As mentioned above: There is even no information about the date of sampling. 1) When were the groundwater samples (191) collected? 2) How many replicates were made? Was the concentrations of hydrochemical and physical parameters were monitored in the same year/period?

Response 5: Thanks for the reviewer’s kind warning. The water samples were collected once in 1999. The revised details can be found in section of “Sample collection and analysis”.

Point 6: What methods were used to develop Figures 5 and 6. Which classification (interpolation) technique was used?

Response 6: The Figure 5 and 6 were plotted by Surfer 15 and Coreldraw X8. The interpolation technique is Kriging method.

Point 7: In addition, it is worth showing a scale bar on both figures (instead of a north arrow). Figure 6 is illegible, please reorganize map composition. Legend entry “County” is not evident - can you present counties with point feature type?

Response 7: Thanks for the reviewer’s kind suggestion. The Figure 5 and 6 have been redrawn. The revised details can be found in Figure 5 and 6.

Reviewer 2 Report

Assessment of Groundwater Quality and Human Health Risk (HHR) Evaluation of Nitrate in the Central-Western Guanzhong Basin, China

Comments:

In this study, 191 groundwater samples were collected to analyze the quality of groundwater and its influence to human health. They analyzed the cations and anions abundance in groundwater, especially the nitrate distribution. For HHR assessment, their result showed that the 6- to 12-month age groups were more likely to suffer from health complications with a larger concentration of nitrate, followed by 6- to 11- year, 21- to 65- year, 18- to 21- year, older than 65- year, 11- to 16- year, and 16- to 18- year  age groups which mainly related to the discrepancy of exposure parameters. This study has great meaning groundwater protection and human health.

The manuscript is in general well written, but this reviewer recommends the authors to revise it. While significant results and straightforward discussion is provided, indicating this work brings major contribution to the field, there are some comments and corrections below, which this reviewer recommends to be addressed by the authors.

Major comments:

The age range, 21- to 65- year is too big. Could you separate them into 3 to 4 groups? What is the meaning to test HCO3- in the water? How about the SO42− influence to human health? 6 the cities and counties marked in black are hard to see, could you change them to other colors? If you combined the industry types in different area with your findings, you can interpret your data more deeply.  The written still need to be polished.

Author Response

Dear reviewer:

Thank you so much for giving us warning and suggestion to revise our manuscript. We have considered comments from you carefully and have made revisions to address your concerns. It is worth noting that all revised details are marked in blue.

Sincerely,

Hui Qian

Point 1: The age range, 21- to 65- is too big. Could you separate them into 3 to 4 groups?

Response 1: Thanks for the reviewer’s kind suggestion. People from 21 to 65 years of age, the body's height, shape and other characteristics have been fully developed, and will not have major changes, so it is more representative in the process of human health risk assessment. And I checked a lot of literature and didn't find the relevant calculation parameters. I am very sorry about it.

Point 2: What is the meaning to test HCO3- in the water?

Response 2: The detection of bicarbonate can be used to analyze the type and formation mechanism of hydrochemistry of water, and is one of the main eight major ions (Na+, K+, Ca2+, Mg2+, Cl-, SO42-, HCO3-, and CO32-). The details can be found in the part of “Groundwater Chemistry”.

Point 3: How about the SO42- influence to human health?

Response 3: Sulfate ion is harmless to the human body and has a great effect on maintaining osmotic pressure in the human body. However, the combination of sulfate ions and other ions can cause harm to the human body. For example, sodium sulfate is irritating to the eyes and skin and Sulfate and calcium ions tend to form precipitates.

Point 4: The cities and counties marked in black are hard to see, could you change them to other colors?

Response 4: Thanks for the reviewer’s kind suggestion. We have changed the color of cities and counties in all figures. The revised details can be found in Figure 1, 5 and 6.

Point 5: If you combined the industry types in different area with your findings, you can interpret your data more deeply?

Response 5: Thanks for the reviewer’s kind suggestion. We have added the impact of industry on groundwater quality. The revised details can be found in the section of “Distribution and Occurrence of Nitrate”.

Point 6: The written still need to be polished

Response 6: Thanks for the reviewer’s kind suggestion. Our manuscript has been revised by a professional English editing service.

Reviewer 3 Report

The paper by Zhang et al. with title: "Assessment of Groundwater Quality and Human Health Risk (HHR) Evaluation of Nitrate in the Central-Western Guanzhong Basin, China" provides an interesting research of nitrate pollution and the impact to population. The authors used groundwater chemical analysis, water quality index and Human health risk (HHR) to estimate the groundwater conditions in the study area.

Comments:

Figure 1: Add coordinate system and convert “sample location” to “groundwater samples”. Figure 5: the image is not representative. There are no groundwater samples in the NE part of the map, so have to change the extension of WQI colors. A geological map can endorse the chemical result, because dolomites referred in text. Provide future works to update the concept (ex. add surface water samples). How nitrate pollution can be treated? Add some management strategies. Add ionic ratios to enhance water source determination results. Add a table with the results of the mentioned studies in the region or add a new figure referring to them. As only groundwater samples collected, add information about aquifers types and their characteristics. Add some information about population (ex. variation of population during the years).

The discussion section should be enforces incorporating methods for multi process assessment (e.g. Statistical analysis) and protection of groundwater (e.g. the concept of vulnerability)  

Suggested literature:

Machiwal D., Cloutier V., Güler C., Kazakis N. (2018) A Review of GIS-Integrated Statistical Techniques for Groundwater Quality Evaluation and Protection. Environmental Earth Science. 77:681.

Author Response

Dear reviewer:

Thank you so much for giving us warning and suggestion to revise our manuscript. We have considered comments from you carefully and have made revisions to address your concerns. It is worth noting that all revised details are marked in red.

Sincerely,

Hui Qian

Point 1: Figure 1: Add coordinate system and convert “sample location” to “groundwater samples”.

Response1: Thanks for the reviewer’s kind suggestion. The coordinate system has been added in Figure 1, and the word of “sample location” also has converted to the “groundwater samples”. The revised details can be found in Figure 1.

Point 2: Figure 5: the image is not representative. There are no groundwater samples in the NE part of the map, so have to change the extension of WQI colors.

Response 2: Thanks for the reviewer’s kind warning. The WQI color in the NE part of the map has changed. The revised details can be found in Figure 5.

Point 3: A geological map can endorse the chemical result, because dolomites referred in text.

Response 3: Thanks for the reviewer’s kind suggestion. We have added the geological map of study area. The revised details can be found in Figure 1(b).

Point 4: Provide future works to update the concept (ex. add surface water samples).

Response 4: Thanks for the reviewer’s kind suggestion. The future works have been provided to update the concept in the paper. The revised details can be found in the part of “Sustainable Groundwater Quality Management”.

Point 5: How nitrate pollution can be treated? Add some management strategies.

Response 5: Thanks for the reviewer’s kind suggestion. Some management strategies have been added in this manuscript. The revised details can be found in the part of “Sustainable Groundwater Quality Management”.

Point 6: Add ionic ratios to enhance water source determination results.

Response 6: Thanks for the reviewer’s kind suggestion. The ionic ratios can be represented by the bivariate diagram and we had written this in the part of “Mechanisms of Hydeogeochemistry” of this manuscript. Very thanks again for the reviewer’s kind suggestion.

Point 7: Add a table with the results of the mentioned studies in the region or add a new figure referring to them.

Response 7: Thanks for the reviewer’s kind suggestion. We have added a new figure referring to the results of non-carcinogenic health risk (the values of HI) for different age groups. The revised details can be found in the Figure 7 of the revised draft.

Point 8: As only groundwater samples collected, add information about aquifers types and their characteristics.

Response 8: Thanks for the reviewer’s kind suggestion. The information about aquifers types and their characteristics have been added in this manuscript. The revised details can be found in the part of “Study Area”.

Point 9: Add some information about population (ex. variation of population during the years).

Response 9: Thanks for the reviewer’s kind suggestion. The population information has been introduced in detail. The revised details can be found in the part of “Study Area”.

Point 10: The discussion section should be enforces incorporating methods for multi process assessment (e.g. Statistical analysis) and protection of groundwater (e.g. the concept of vulnerability).

Suggested literature:

Machiwal D., Cloutier V., Güler C., Kazakis N. (2018) A Review of GIS-Integrated Statistical Techniques for Groundwater Quality Evaluation and Protection. Environmental Earth Science. 77:681.

Response 10: Thanks for the reviewer’s kind suggestion. I have carefully read the article you mentioned. The statistical analysis is indeed a good method to process data, but the main research object of this paper is nitrogen. Whether it is cluster analysis (CA) or principal component analysis (PCA), it is not suitable in this paper. Therefore, there is no choice of statistical analysis. However, for the methods for protection of groundwater, we have revised it. The revised details can be found in the sections of “Sustainable Groundwater Quality Management”.

Reviewer 4 Report

Manuscript ijerph-615476 is an interesting study and includes clear descriptions and explanations of the experimental design. I see great authors’ expertise in the water geochemistry and the paper could be interesting for readers. Following comments aim to improve the quality of the manuscript:

- Manuscript has some grammar and language issues, which need to be addressed. Please, careful review of the text.

- Please, increase the readability of figures 5 and 6

- The introduction must be rewritten. It looks like a collage of sentences taken from other papers. For example, the sentence: “evaluated the groundwater contamination for fluoride and nitrate in semi-arid region of Nirmal Province, South India and found that ingestion of high fluoride and nitrate water could be the chief reason for health risk” was copied from Adimalla, N., Li, P., & Qian, H. (2019). Evaluation of groundwater contamination for fluoride and nitrate in semi-arid region of Nirmal Province, South India: a special emphasis on human health risk assessment (HHRA). Human and Ecological Risk Assessment: An International Journal, 25(5), 1107-1124.

- The manuscript must take into consideration the data and conclusions proposed by Li, Peiyue, et al. "Geochemistry, hydraulic connectivity and quality appraisal of multilayered groundwater in the Hongdunzi Coal Mine, Northwest China." Mine Water and the Environment 37.2 (2018): 222-237.

Author Response

Dear reviewer:

Thank you so much for giving us warning and suggestion to revise our manuscript. We have considered comments from you carefully and have made revisions to address your concerns. It is worth noting that all revised details are marked in purple.

Sincerely,

Hui Qian

Point 1: Manuscript has some grammar and language issues, which need to be addressed. Please, careful review of the text?

Response 1: Thanks for the reviewer’s kind suggestion. Our manuscript has been revised by a professional English editing service.

Point 2: Please, increase the readability of figures 5 and 6

Response 2: Thanks for the reviewer’s kind suggestion. The Figure 5 and 6 have been redrawn. The revised details can be found in Figure 5 and 6.

Point 3: The introduction must be rewritten. It looks like a collage of sentences taken from other papers. For example, the sentence: “evaluated the groundwater contamination for fluoride and nitrate in semi-arid region of Nirmal Province, South India and found that ingestion of high fluoride and nitrate water could be the chief reason for health risk” was copied from Adimalla, N., Li, P., & Qian, H. (2019). Evaluation of groundwater contamination for fluoride and nitrate in semi-arid region of Nirmal Province, South India: a special emphasis on human health risk assessment (HHRA). Human and Ecological Risk Assessment: An International Journal, 25(5), 1107-1124.

Response 3: Thanks for the reviewer’s kind warning. I have rewritten the introduction of this manuscript. The revised details can be found in the part of “Introduction”.

Point 4: The manuscript must take into consideration the data and conclusions proposed by Li, Peiyue, et al. “Geochemistry, hydraulic connectivity and quality appraisal of multilayered groundwater in the Hongdunzi Coal Mine, Northwest China." Mine Water and the Environment 37.2 (2018): 222-237.

Response 4: Thanks for the reviewer’s kind suggestion. I had carefully read the article you mentioned, which focused on geochemistry, hydraulic connectivity and quality appraisal of unconfined aquifer and upper confined aquifer in a part of Yinchuan Plain. In addition, the methods and conclusions of this article are worth learning. Therefore, the article is cited as reference in our manuscript. The details can be found in the manuscript.

Round 2

Reviewer 3 Report

I have read the revised manuscript and i have no further comments